# Muscle and Muscle-like Autoantigen Expression in Myasthenia Gravis Thymus: Possible Molecular Hint for Autosensitization

**DOI:** 10.3390/biomedicines11030732

**Published:** 2023-02-28

**Authors:** Nicola Iacomino, Letizia Scandiffio, Fabio Conforti, Erika Salvi, Maria Cristina Tarasco, Federica Bortone, Stefania Marcuzzo, Ornella Simoncini, Francesca Andreetta, Daniela Pistillo, Emanuele Voulaz, Marco Alloisio, Carlo Antozzi, Renato Mantegazza, Tommaso Martino De Pas, Paola Cavalcante

**Affiliations:** 1Neurology IV—Neuroimmunology and Neuromuscolar Diseases Unit, Fondazione IRCCS Istituto Neurologico Carlo Besta, 20133 Milan, Italy; 2Medical Oncology Division, Humanitas Gavazzeni, 24125 Bergamo, Italy; 3Neuroalgology Unit, Fondazione IRCCS Istituto Neurologico Carlo Besta, 20133 Milan, Italy; 4Ph.D. Program in Neuroscience, University of Milano-Bicocca, via Cadore 48, 20900 Monza, Italy; 5Center for Biological Resources, Humanitas Cancer Center, IRCCS Humanitas Research Hospital, 20089 Rozzano, Italy; 6Department of Biomedical Sciences, Humanitas University, 20090 Pieve Emanuele, Italy; 7Thoracic Surgery Division, IRCCS Humanitas Research Hospital, 20089 Rozzano, Italy

**Keywords:** myasthenia gravis, autoimmunity, thymus, thymoma, muscle antigens, muscle-like tumor antigens

## Abstract

The thymus is widely recognized as an immunological niche where autoimmunity against the acetylcholine receptor (AChR) develops in myasthenia gravis (MG) patients, who mostly present thymic hyperplasia and thymoma. Thymoma-associated MG is frequently characterized by autoantibodies to the muscular ryanodine receptor 1 (RYR1) and titin (TTN), along with anti-AChR antibodies. By real-time PCR, we analyzed muscle—CHRNA1, RYR1, and TTN—and muscle-like—NEFM, RYR3 and HSP60—autoantigen gene expression in MG thymuses with hyperplasia and thymoma, normal thymuses and non-MG thymomas, to check for molecular changes potentially leading to an altered antigen presentation and autoreactivity. We found that *CHRNA1* (AChR-α subunit) and *AIRE* (autoimmune regulator) genes were expressed at lower levels in hyperplastic and thymoma MG compared to the control thymuses, and that the *RYR1* and *TTN* levels were decreased in MG versus the non-MG thymomas. Genes encoding autoantigens that share epitopes with AChR-α (NEFM and HSP60), RYR1 (neuronal RYR3), and TTN (NEFM) were up-regulated in thymomas versus hyperplastic and control thymuses, with distinct molecular patterns across the thymoma histotypes that could be relevant for autoimmunity development. Our findings support the idea that altered muscle autoantigen expression, related with hyperplastic and neoplastic changes, may favor autosensitization in the MG thymus, and that molecular mimicry involving tumor-related muscle-like proteins may be a mechanism that makes thymoma prone to developing MG.

## 1. Introduction

Myasthenia gravis (MG) is an autoimmune disorder affecting the neuromuscular junction (NMJ). The predominant clinical manifestations are fluctuating skeletal muscle weakness and fatigability. In most MG patients (~85%), the autoimmune attack is against the nicotinic acetylcholine receptor (AChR), which is located at the post-synaptic endplate; additional autoantigens are the muscle-specific tyrosine kinase (MuSK, 5–8% of patients) and the lipoprotein-related protein 4 (LRP4, 1–33% of patients), which are two proteins that are involved in AChR clustering at the NMJ [1,2]. Anti-AChR, anti-MuSK, and anti-LRP4 antibodies are pathogenetic, leading to post-synaptic membrane damage and the impairment of neuromuscular transmission [2]. A proportion of MG patients are considered “triple seronegative”, as they have no detectable antibodies to the known autoantigens [2]. 

The thymus has been recognized as the prime site of autosensitization in AChR-MG patients, who mostly (~80%) present thymic abnormalities, including follicular hyperplasia and thymoma [3,4]. Its role as an immunological niche for autoimmunity development and perpetuation in MG patients is supported by the beneficial effects of thymectomy in patients with a hyperplastic thymus, as demonstrated by the MGTX trial and its 2-year extension study [5,6]. Thymic hyperplasia is present in about 60% of AChR-MG patients, who are mostly female with an early onset (<50 years) disease [3]. Hyperplastic thymus is characterized by medullary B-cell infiltrates organized in ectopic germinal centers (GCs) forming follicles, which are surrounded by plasma cells and muscle-like myoid cells [7,8,9]. Myoid cells are suspected to participate in the loss of tolerance to muscle autoantigens since they do not express HLA-class II molecules but they express AChR in its native conformation, along with other muscle cell proteins, including the muscular ryanodine receptor 1 (RYR1) and titin (TTN), whose fragments could be abnormally released and taken up by dendritic cells to be cross-presented to autoreactive T cells [10,11,12,13]. Thymic epithelial cells (TECs) do not express functional AChR [14] but they express HLA-class II molecules and potentially immunogenic AChR subunits [15]. Medullary TECs (mTECs) are essential for central tolerance establishment, as they express peripheral tissue-restricted self-antigens (TRAs) and induce self-reactive T-cell deletion through direct antigen presentation or cross-presentation by dendritic cells [16,17]. Self-antigen expression in mTECs occurs via the autoimmune regulator (AIRE) transcription factor, which controls the transcriptional levels of *CHRNA1*, the gene encoding the AChR α-subunit (AChR-α) containing the main immunogenic region [18]. 

Thymoma is a thymic epithelial tumor that is frequently associated with autoimmune disorders, mainly MG, in up to 30–45% of thymoma patients [19,20]. Conversely, up to 30% of MG patients present a thymoma [21]. The World Health Organization (WHO) classification distinguishes type A, AB, B1, B2, and B3 thymomas, depending on the epithelial cell features and lymphocyte content [22,23]. Type B2 thymoma is the most common type observed in MG, followed by AB and B1 [19,21]. Thymoma-associated MG can occur at any age, more frequently in late-onset (>50 years) MG patients. The disease is usually more severe and less responsive to treatment than non-thymomatous MG, and it is frequently associated with autoantibodies to RYR1 and TTN, along with anti-AChR antibodies [4,24]. Defective expression of AIRE and HLA-class II molecules, reduction or absence of myoid cells, abnormal T-cell selection, and the failure in regulatory T-cell (Treg) generation are all pathogenic features of thymoma that increase susceptibility to autoimmunity, and particularly to MG, in thymoma patients [24].

The expression of muscle autoantigens in the pathological MG thymus has been extensively investigated to check for changes leading to an altered antigen presentation and autoreactivity, but conclusive results are lacking [15,25,26]. Of interest, microarray profiling performed by Radovich and colleagues [27] revealed an intra-tumoral overexpression of genes encoding for muscle autoantigens, and antigens with a sequence similarity to muscle autoantigens (“muscle-like antigens”), in MG with thymoma. Indeed, (i) CHRNA1 was 3.0-fold more expressed in MG than non-MG thymomas; (ii) expression levels of the medium-sized neurofilament (NEFM), which exhibits immunogenic similarities with AChR-α and TTN, showed a 23.8-fold increase in MG compared to non-MG thymomas; and (iii) the main neuronal RYR3, which shares homology with muscular RYR1 and cardiac RYR2, was also up-regulated (5.5-fold) in MG thymomas [27]. The TTN and AChR-α epitope presence in NEFM was previously described by Marx and colleagues [28,29], suggesting molecular mimicry as a putative mechanism for anti-AChR and anti-TTN autosensitization in thymomatous MG patients. HSP60 has been described as an additional muscle-like antigen, since it also shares immunogenic epitopes with AChR-α [30].

To further address the question whether the intra-thymic expression of muscle autoantigens is altered in AChR-MG patients, we searched for possible changes in the *CHRNA1*, *RYR1*, and *TTN* expression levels, in relationship with *AIRE* expression, in hyperplastic MG thymuses and MG thymomas. Moreover, to explore the molecular mimicry hypothesis, we studied the expression of *NEFM*, *RYR3*, and *HSP60* in the same tissues.

Our findings reveal molecular differences among hyperplastic, thymomatous, and control thymuses, suggesting a role for muscle and muscle-like autoantigens as possible molecular helpers for the development of an autoimmune response in the MG thymus. We also observed histotype-specific alterations of muscle and muscle-like autoantigen expression profile in thymomas, which could be implicated in thymoma-associated MG.

## 2. Results

### 2.1. Altered Expression of Muscle Autoantigens in Hyperplastic and Thymoma MG Thymuses

Transcriptional levels of *CHRNA1*, *RYR1*, and *TTN* autoantigen genes were assessed in hyperplastic and thymoma thymuses of MG patients, non-MG thymomas, and normal thymuses (Table 1) using real-time PCR.

According to the WHO histological classification, MG and non-MG thymomas were grouped as follows: type A/AB, including A and AB thymomas; type B1/B2, including B1, B2, and B1/B2 mixed thymomas; and type B3/B3 mixed, including B3 and B2/B3 mixed thymomas.

The *CHRNA1* mRNA levels were unchanged in hyperplastic MG compared to the normal control thymuses when patients were not stratified according with pre-thymectomy treatment (Figure 1a), but they were significantly lower in hyperplastic thymuses from corticosteroid-naïve patients compared to the control thymuses and hyperplastic thymuses from corticosteroid-treated patients (Figure 1b). A significant reduction in *CHRNA1* expression was also observed in MG thymomas, particularly in thymomas from corticosteroid-naïve patients, compared to non-MG thymomas and control thymuses (Figure 1a,c). This reduction was mainly observed in the MG type B (B1/B2 and B3/B3 mixed) subset compared to the same histological subset in the non-MG thymoma group (Figure 1d). 

*RYR1* expression was significantly reduced in hyperplastic and thymoma thymuses from corticosteroid-naïve, but not corticosteroid-treated, MG patients compared to control thymuses and non-MG thymomas, respectively (Figure 2a–c). In MG thymomas, *RYR1* down-regulation mainly involved type A/AB tumor subset compared to the corresponding subset in non-MG tumors (Figure 2d). In both MG and non-MG thymomas, type A/AB tumors showed higher *RYR1* mRNA levels than the histological B subtypes (Figure 2d). 

The *TTN* mRNA levels were also significantly lower in MG than non-MG thymomas, but not in hyperplastic MG compared to the control thymuses, with no differences observed between the corticosteroid-naïve and -treated patients (Figure 2e–g). As observed for *RYR1*, the type A/AB subset showed lower *TTN* mRNA levels in MG compared to non-MG thymomas, but the differences were not statistically significant (Figure 2h). A slight, but not significant reduction in the *TTN* mRNA levels was also observed in the MG B3/B3 mixed thymoma subtype compared to the same subtype in non-MG thymomas (Figure 2h).

We did not find any differences in the *CHRNA1*, *RYR1*, and *TTN* expression levels between male and female patients, nor between normal thymuses from cardiopathic subjects and normal thymuses adjacent to thymoma in non-MG thymoma patients used as the controls (data not shown).

### 2.2. Decreased Expression of AIRE in MG Thymomas

*AIRE* expression was assessed in hyperplastic and thymoma MG thymuses, non-MG thymomas, and normal thymuses (Table 1), using real-time PCR. We found that the *AIRE* mRNA levels were not altered in hyperplastic MG compared to control thymuses (Figure 3a,b), but they were significantly lower in MG thymomas, for both corticosteroid-naïve and -treated patients, compared to non-MG thymomas, control thymuses, and hyperplastic MG thymuses (Figure 3a,c).

Reduction in *AIRE* in MG thymomas was mainly observed in the type A/AB and B1/B2 subsets compared to the corresponding subsets in the non-MG thymoma group (Figure 3d). The *AIRE* mRNA levels positively correlated with those of *CHRNA1* in hyperplastic thymuses and thymomas from untreated MG patients (Pearson r = 0.83 and *p* < 0.0001; data not shown). In both the MG and non-MG thymoma groups, *AIRE* was under-expressed in the type B3/B3 mixed subset compared to the type B1/B2 subset (Figure 3d). No difference was found in the *AIRE* expression between male and female MG patients, nor between normal thymuses from cardiopathic subjects and those adjacent to thymoma in non-MG thymoma patients (data not shown). We observed a slightly reduced *AIRE* expression in the normal thymuses of female compared to the male control subjects (data not shown), which is in line with previous data on lower *AIRE* levels in females than males [31]. 

### 2.3. NEFM, RYR3, and HSP60 as Potential Muscle-like Autoantigens Underlying Molecular Mimicry in MG Thymus

Based on the previously reported similarity of NEFM with AChR-α and TTN, of RYR3 with RYR1, and of HSP60 with AChR-α [27,28,29,30], we addressed the hypothesis that the altered expression of these muscle-like autoantigens in the hyperplastic and neoplastic MG thymic microenvironment could underlie the intra-thymic autosensitization phenomena toward muscle antigens via molecular mimicry. We thus assessed the *NEFM*, *RYR3*, and *HSP60* expression in hyperplastic and thymoma MG thymuses, non-MG thymomas, and normal thymuses (Table 1), using real-time PCR. 

Our investigation revealed a lower expression of *NEFM* in hyperplastic MG thymuses from corticosteroid-naïve patients compared to control thymuses and hyperplastic thymuses from corticosteroid-treated patients, who showed increased thymic *NEFM* mRNA levels (Figure 4a,b). Of note, the *NEFM* expression was up-regulated in both MG and non-MG thymomas compared to hyperplastic MG and control thymuses (Figure 4a), with no significant difference observed between the two thymoma groups, as well as between thymomas from corticosteroid-naïve and -treated MG patients (Figure 4c). However, by stratifying the tumors according to the WHO type, we identified a significant *NEFM* mRNA level increase in type A/AB MG thymomas compared to the corresponding subset in the non-MG thymoma group (Figure 4d). 

*RYR3* expression showed a trend similar to that of *NEFM* in hyperplastic MG thymuses: it was reduced in the thymus from corticosteroid-naïve patients, and up-regulated in corticosteroid-treated patients, compared to control thymuses (Figure 4e,f). Interestingly, a significant *RYR3* overexpression was observed in MG thymomas, from both corticosteroid-naïve and -treated patients, compared to the non-MG thymomas, control thymuses, and hyperplastic MG thymuses (Figure 4e,g). This overexpression was consistent across the WHO subtypes but was mainly involved the A/AB and the B3/B3 mixed subtypes compared to the corresponding subtypes in non-MG thymomas (Figure 4h). 

The HSP60 mRNA levels did not differ between hyperplastic thymuses from untreated MG patients compared to control thymuses, but they were significantly increased in hyperplastic thymuses from patients treated with corticosteroid before thymectomy, which was likely as an effect of treatment (Figure 5a,b). An effect of corticosteroid treatment on *HSP60* expression was also observed in MG thymomas, since the *HSP60* transcriptional levels were higher in corticosteroid-treated compared to -naïve MG thymoma patients (Figure 5c). As observed for *NEFM*, both MG and non-MG thymomas showed higher *HSP60* expression levels compared to hyperplastic MG and control thymuses (Figure 5a), which was likely related to thymic neoplastic changes. These levels were lower in the MG thymoma, particularly of type A/AB, compared to non-MG thymomas (Figure 5a,c,d). To address the hypothesis of “HSP60-AChR-α cross-reactivity” [30] in MG patients, we assessed the presence of anti-Hsp60 IgG antibodies in pre-thymectomy sera from AChR-MG patients with thymic follicular hyperplasia and thymomas, seronegative MG patients, and healthy controls. We did not find differences among the MG patients and control groups (data not shown), thus indicating a lack of association between antibodies to HSP60 and the disease in our cohort of patients. 

We did not find differences in the *NEFM, RYR3*, and *HSP60* expression levels between normal thymuses from cardiopathic subjects and normal thymuses adjacent to thymoma in non-MG thymoma patients (data not shown).

### 2.4. Heatmap of Muscle and Muscle-like Autoantigenes in MG and Control Thymuses

We performed a heatmap analysis of our overall molecular data to visualize the relationship among the investigated genes based on their differential expression in MG compared to control thymuses. We revealed two separate gene patterns: the first included the muscle autoantigen *CHRNA1* and *RYR1* genes, but not *TTN*, clustered together in one group along with *AIRE*; the second pattern included the muscle-like autoantigen genes, *NEFM*, *RYR3*, and *HSP60*, along with *TTN* (Figure 6). Of note, the genes encoding muscle autoantigens showed a lower expression in MG (both hyperplastic thymuses and thymomas) compared to normal thymuses and non-MG thymomas, suggesting their down-regulation as a possible molecular hint for intra-thymic autosensitization. Contrariwise, genes encoding muscle-like autoantigens showed a trend of increasing in the thymomas compared to normal and hyperplastic thymuses (Figure 6), suggesting their abnormal overexpression as a molecular event that makes the thymoma environment prone to autosensitization toward muscle antigens. *CHRNA1* and *AIRE* were clustered together in one group, which is in line with the *CHRNA1* regulation by AIRE [18]. Among the genes encoding muscle-like autoantigens, *NEFM* and *RYR3* had a more similar gene expression pattern and were clustered together in one group; they were both particularly overexpressed in type A/AB MG thymomas compared to the same tumor type in the non-MG thymoma group (Figure 6). 

### 2.5. Autoantibody Status and Intra-Thymic Muscle and Muscle-like Autoantigen Expression

All MG patients with thymic hyperplasia and thymoma who were included in the study were positive for anti-AChR antibodies. The autoantibody titer did not correlate with the expression levels of *CHRNA1, NEFM,* and *HSP60* in follicular hyperplastic and thymoma MG thymuses. Anti-RYR1 and -TTN antibodies were absent in patients with hyperplastic thymuses, which was as expected. Anti-RYR1 antibodies were detected in available sera from 13 of 17 MG patients with thymoma, including 6 out of 6 patients with type A/AB, 4 out of 5 patients with type B1/B2, and 3 out of 6 patients with type B3/B3 mixed thymoma. Anti-TTN antibodies were detected in the serum from 9 out of 17 thymoma MG patients, including 4 out of 6 patients with type A/AB, 2 out of 5 patients with type B1/B2, and 3 out of 6 patients with type B3/B3 mixed thymoma. 

Anti-AChR, anti-RYR1, and anti-TTN antibodies were tested in 13 available sera from non-MG thymoma patients, including 7 patients with type A/AB, 5 patients with type B1/B2, and 1 patient with type B3/B3 mixed thymoma. Among them, two patients were positive for anti-AChR antibodies: one with type A/AB and one with type B1/B2 thymoma. One patient with type A/AB thymoma, who was negative for anti-AChR antibodies, was positive for both anti-RYR1 and anti-TTN antibodies; the remaining non-MG thymoma patients were negative for autoantibodies. 

The small sample size of the histological tumor subgroups did not allow us to draw definitive conclusions on the relationship between muscle/muscle-like autoantigen expression and autoantibody (anti-RYR1 and anti-TTN) profile in thymoma patients. However, by comparing anti-RYR and anti-TTN antibody-positive MG patientswith seronegative non-MG thymoma patients, we observed that MG patients with type A/AB thymoma, who were all positive for anti-RYR1 antibodies, had a lower intra-thymoma *RYR1* expression, associated with *AIRE* down-regulation, compared to seronegative non-MG patients with thymoma of the same WHO type (Figure 7a). 

*RYR1* was also down-regulated in thymomas from anti-RYR1 antibody-positive MG patients with B3/B3 mixed thymomas, who showed a down-regulation of *AIRE* and an up-regulation of *RYR3*, compared to thymomas of the same type in seronegative non-MG patients (Figure 7a). Among the anti-TTN antibody-positive MG patients, we observed the following: patients with type A/AB thymomas had lower *TTN* and *AIRE* expression levels, which were associated with higher levels of *NEFM*; patients with type B1/B2 thymoma also had lower levels of *TTN* and *AIRE*; and patients with B3/B3 mixed thymomas had higher levels of *NEFM*, which was associated with the *AIRE* decrease, compared to seronegative non-MG patients having the same thymoma type (Figure 7b).

## 3. Discussion

The thymus is a complex and specialized primary lymphoid organ that is essential for the maturation and differentiation of T cells, and for the establishment of central self-tolerance. This process occurs during T-cell migration within the cortical and medullary thymic compartments, where low or high affinity interactions of immature cells with self-antigens displayed by mTECs, myoid cells, and dendritic cells, lead to positive and negative T-cell selection [32]. Expression of TRAs in mTECs, which is crucial for the deletion of autoreactive T cells, is regulated by the autoimmune regulator AIRE. AIRE is a critical transcription factor for shaping immunological tolerance and preventing autoimmunity, and its pathogenic variants are associated with autoimmune diseases [33].

It is now widely accepted that the thymus is the main site of autoimmunity development and perpetuation in AChR-MG patients, who mostly present thymic follicular hyperplasia or thymoma. The MG thymus contains all the elements required for triggering and sustaining an autoimmune reaction against the AChR, including the autoantigen itself (i.e., AChR/AChR subunits), expressed in muscle-like myoid cells and TECs, and autoreactive T and B cells [34]. Myoid cells and TECs also express other muscle antigens, including RYR1 and TTN, which are two autoantigens in thymoma-associated MG [10,11,12,13]. Immunological alterations that are associated with hyperplasia and thymoma make the thymus susceptible to autoimmunity: (i) chronic inflammation, GC formation, and hyper-activation of innate immune responses, which are likely due to persistent pathogen infections (i.e., Epstein–Barr virus), are pathogenetic features favoring autoimmunity development in hyperplastic thymus [34,35,36,37]; and (ii) defective AIRE and HLA-class II molecule expression, myoid cell reduction, and the failure of Treg differentiation are pathogenetic features favoring autoreactivity in thymoma [24]. Nevertheless, the exact molecular mechanisms leading to specific autosensitization toward AChR, or to RYR1 and TTN, in the thymus of MG patients are not completely known. Since intra-thymic autoantigen expression plays a key role in self-tolerance maintenance/breakdown, altered AChR/muscle antigen expression into the thymus could favor autoreactivity in a genetically susceptible background, or in a microenvironment prone to autoimmunity. A close link between AChR expression and autosensitization in the inflamed hyperplastic MG thymic milieu has been postulated, since pro-inflammatory and Toll-like receptor (i.e., TLR3) stimuli are able to increase the AChR-α subunit expression in TECs in vitro, and to induce thymic hyperplastic changes and anti-AChR autoantibody production in vivo [38]. However, whether the expression of muscle autoantigens is altered in MG thymuses, hence contributing to intra-thymic autoreactivity to NMJ components, is still an unsolved issue. 

Several early studies have searched for evidence of an altered expression of the AChR-α subunit in MG thymuses; RYR1 and TTN expression was also widely investigated in MG thymomas. Liu and colleagues [25] reported a lower *CHRNA1* and *AIRE* expression in MG thymomas, mainly of types B2 and B3, compared to non-MG thymomas. RYR1 and TTN epitopes, along with costimulatory molecules, were detected in neoplastic thymoma cells, suggesting a primary autosensitization to these autoantigens in the neoplastic thymic tissue [26]. An overexpression of thymic muscle-like epitopes, including neurofilaments, was identified in thymomas, suggesting molecular mimicry involving tumor antigens as a mechanism promoting T-cell immunization to muscle autoantigens [28]. More recently, using microarray profiling, Guo and colleagues [39] demonstrated lower transcriptional levels of *CHRNA3* and *AIRE* in MG compared to non-MG thymomas. Interestingly, by multi-platform omics analyses, Radovich and colleagues [27] revealed tumoral overexpression of genes encoding muscle and muscle-like autoantigens, including NEFM (mimicking AChR-α and TTN) and RYR3 (mimicking RYR1), in MG compared to non-MG thymomas, also showing molecular differences across WHO types that could be of pathogenic relevance for thymoma-associated MG. These authors provided new insights into the molecular link between thymoma and MG, although experimental validation of the microarray data was not performed. 

Less attention has been devoted in the past to the expression of muscle and muscle-like autoantigens in follicular hyperplastic MG thymuses. However, Wakkach and colleagues [15] showed similar *CHRNA1* mRNA levels in thymic suspensions and TECs from non-thymomatous MG patients and control subjects using PCR.

To gain further evidence on intra-thymic autoantigen alterations that are potentially associated with patients’ risk of developing MG, we performed a comprehensive molecular analysis of muscle—*CHRNA1*, *RYR1*, and *TTN*—and muscle-like—*NEFM*, *RYR3*, and *HSP60*—autoantigen gene expression in both follicular hyperplastic MG and thymoma thymuses, in comparison with normal control thymuses and thymomas from non-MG patients. The expression of *HSP60,* along with *NEFM*, was examined in view of the reported *HSP60* similarities with AChR-α [30]. Of interest, we found lower expression levels of *CHRNA1* in both hyperplastic and thymoma MG thymuses, particularly type B1/B2 MG thymomas, compared to normal thymuses and non-MG thymomas. These data were in line with those of Liu and colleagues [25], but not with the results by Radovich and colleagues, who showed *CHRNA1* overexpression in MG versus non-MG thymomas [27]. Since *CHRNA1* expression is controlled by AIRE [18], which is defective in thymomas [24], a reduction in *CHRNA1* expression rather than an increase in these pathological tissues could be expected. Accordingly, lower *CHRNA1* mRNA levels were associated with significantly reduced *AIRE* mRNA levels in our MG thymoma samples, and with slightly reduced *AIRE* mRNA levels in hyperplastic MG thymuses. Indeed, we found a positive correlation between *CHRNA1* and *AIRE* expression levels in both hyperplastic thymuses and thymomas. The reduced *CHRNA1* expression we observed in MG thymuses was also in agreement with data showing that the biallelic variant rs16862847, which is associated with early-onset MG, is able to prevent the binding of the interferon regulatory factor 8 to the promoter of *CHRNA1* and abrogate the promoter activity in TECs in vitro [18]. Very recently, a genome-wide association study (GWAS), combined with a transcriptome-wide association study (TWAS) using expression data from skeletal muscles, whole blood, and tibial nerve revealed that a disease-associated variant, located within the CCAAT–enhancer-binding protein beta transcription factor binding site of *CHRNA1* and within the antisense gene to *CHRNA1* (AC010894.2) on the reverse strand, may decrease *CHRNA1* expression [40]. 

Defective *AIRE* expression in neoplastic epithelial cells is well-known [24]. Here, we provide results that suggest that this defect is more prominent in MG compared to non-MG thymomas, particularly in type B1/B2 tumors, with B2 being the histology type that is more frequently associated with MG [19,20,21]. Taken together, these data suggest that a lower *CHRNA1* and *AIRE* expression in MG thymuses, particularly thymomas, could signify relevant molecular alterations contributing to anti-AChR autosensitization. Indeed, since intra-thymic autoantigen expression is crucial for the establishment and maintenance of self-tolerance by the negative selection of T cells and differentiation of protective Tregs [32,33], reduced AChR-α expression can impair the negative selection of autoreactive T cells and/or contribute to the deficiency of AChR-α-specific Tregs, in turn favoring autoimmunity [24]. Of note, reduced intra-thymic *CHRNA1*, as well as *AIRE*, mRNA levels were observed in both corticosteroid-naïve and -treated MG thymoma patients, but only in corticosteroid-naïve patients with follicular hyperplastic thymuses. This observation suggests that *CHRNA1* under-expression may be a histopathological feature related to follicular hyperplastic changes, which can be reversed by the corticosteroid treatment. Indeed, glucocorticoids are able to reduce the GC number in MG thymuses and induce molecular changes that are likely related to the acetylation molecular process, which is essential for gene regulation and is over-represented in corticosteroid-treated patients [41].

Similarly to *CHRNA1*, genes encoding for the two thymoma-associated autoantigens RYR1 and TTN were expressed at lower levels in MG compared to non-MG thymomas, with RYR1 being mainly down-regulated in the MG A/AB subset, and TTN in the MG A/AB and B3/B3 mixed subset.

Microarrays data from Radovich and colleagues [27] demonstrated that the transcriptional levels of the medium-sized neurofilament, NEFM, which exhibit immunogenic similarities with AChR-α and TTN, were higher in MG compared to non-MG thymomas, and even higher in the MG A/AB subset. Our real-time PCR analyses showed an *NEFM* overexpression in both MG and non-MG thymomas compared to follicular hyperplastic MG and normal control thymuses. The NEFM expression levels were significantly higher in the MG A/AB subset compared to the same subset in the non-MG thymoma group, which is in line with Radovich’s findings. However, since *NEFM* overexpression was not exclusive to MG thymomas, we think that it may be a histopathological feature of the thymic neoplastic microenvironment that is potentially able to predispose, but not sufficiently, anti-AChR autoimmunity. Additional factors, such as genetic factors (e.g., HLA and other susceptibility loci), are required for MG development. 

The neuronal ryanodine receptor, RYR3, sharing homology with muscular RYR1, was found to be up-regulated in MG compared to non-MG thymomas by Radovich and colleagues [27], with the highest up-regulation being observed in the B1/B2/B3 subset. Accordingly, we found a significant RYR3 overexpression in MG, particularly of type B3/B3 mixed, along with A/AB, compared to non-MG thymomas, hyperplastic MG, and control thymuses, thus suggesting an association between RYR3 overexpression and autoreactivity to RYR1 developing inside the thymoma. 

The increased expression of *HSP60*, which we observed in both MG and non-MG thymomas compared to hyperplastic MG and control thymuses, could be a molecular alteration related to thymic neoplastic transformation. Indeed, HSP60 is overexpressed in a variety of cancers, and it is known to regulate tumor progression and apoptosis [42]. Of note, HSP60 has also been reported as a target self-antigen in autoimmunity, and HSP60-reactive T and B cells have been described as part of immune responses in several infectious diseases [43]. Based on these observations, and considering the previously reported similarity between HSP60 and AChR-α [30], we assessed anti-HSP60 antibodies that were potentially cross-reacting with AChR-α, in AChR-MG patients and healthy controls. We did not find differences between our cohort of MG patients and controls, indicating a lack of association between the anti-HSP60 response and MG. Interestingly, we found an overexpression of *HSP60* in the thymus of corticosteroid-treated compared to -naïve MG patients (both those with hyperplastic thymus and thymoma), suggesting HSP60 induction to be an effect of immunosuppressive treatment. In line with this observation, studies in animal models showed that HSP60, among its multiple roles, also has immune-modulatory functions, and is able to modulate the inflammatory process [43].

Our gene profiling demonstrated heterogeneity in muscle and muscle-like autoantigen expression in the thymomas of different WHO type, reflecting distinct morphology, genetics, and global gene expression profiles of thymomas. Indeed, differences in the expression of functional pathways with immunological relevance among different WHO thymoma types were observed by Yamada and colleagues [44], who suggested that some mechanisms leading to thymoma-associated MG might be different among thymoma histological subtypes. 

A heatmap analysis of our molecular data indicated that muscle autoantigen genes, including *CHRNA1* and *RYR1,* form, along with *AIRE*, a cluster of genes that are down-regulated in MG versus control thymuses, whereas genes encoding muscle-like autoantigens, including *NEFM*, *RYR3*, and *HSP60*, cluster together in a separate pattern of genes that are up-regulated in MG versus non-MG thymomas. These data, along with the gene expression in thymoma patients who are positive for anti-RYR1 and/or -TTN autoantibodies, suggest that the intra-thymic muscle autoantigen down-regulation, and the down-regulation of *AIRE*, are molecular alterations that may favor autosensitization and autoreactivity to muscle autoantigens in MG. On the contrary, the overexpression of genes encoding muscle-like tumor antigens, such as *NEFM*, *HSP60*, and *RYR3*, may be a hallmark of thymic neoplastic tissue, which can predispose to MG in the presence of additional susceptibility molecular/genetic factors (e.g., susceptibility HLA alleles). 

Changes in muscle and muscle-like autoantigen expression in MG thymuses could reflect both gene expression changes in specific thymic cell populations, related to AIRE, and changes in the proportion of specific cells, depending on the pathological thymus. Reduced expression of muscle antigens in MG thymuses could be related to the reduced expression of AIRE in TECs and/or to a reduced number of myoid cells. Since AIRE is known to be expressed in B cells [45], a massive B-cell presence in hyperplastic MG thymuses could affect the global expression of AIRE in these tissues and hide a decrease in TECs. This hypothesis deserves further study, as we are required to understand whether AIRE controls the expression of *RYR1* and/or *TTN*, along with that of *CHRNA1*, or whether other transcription factors, e.g., regulators of TRA expression (e.g., Prdm1, Fezf2) [46,47], underlie the muscle antigen alterations in MG thymuses. NEFM epitopes were previously found to be expressed in thymoma TECs [28]. Of interest, by constructing a comprehensive atlas of thymoma using bulk and single-cell RNA-sequencing, Yasamizu and colleagues [48] recently demonstrated that MG thymomas are characterized by tumor mTECs with an atypical expression profile of neuron-associated molecules, including both NEFM and RYR3, thus strengthening the idea that changes in the expression of these muscle-like neuron-related proteins are due to tumor TECs. Further studies are needed to understand which cell populations abnormally express HSP60 in thymomas.

## 4. Conclusions

The molecular analysis of muscle—CHRNA1, RYR1, and TTN—and muscle-like—NEFM, RYR3, and HSP60—autoantigen gene expression provides new insights in the intra-thymic alterations underlying autosensitization to muscle autoantigens in AChR-MG patients. We suggest that the intra-thymic down-regulation of genes encoding muscle autoantigens (*CHRNA1*, *RYR1*, *TTN*) and *AIRE* may favor autosensitization processes in genetic backgrounds that are susceptible to MG. In the thymoma context, the up-regulation of muscle-like tumor-related autoantigens may contribute to the altered molecular setting that favors autoimmunization. Our data support the idea that the T-cell response to muscle proteins in thymoma patients is favored by gene expression changes that are typical of neoplastic cells, which can explain the association between thymoma and MG. Distinct molecular patterns observed among WHO thymoma types suggest distinct molecular events linking thymoma of different types with MG, implying that the histology stratification of thymoma is an important step when studying thymoma-associated MG pathogenesis, as also highlighted by Yamada and colleagues [44]. 

## 5. Materials and Methods

### 5.1. Patients and Biological Samples

Thymic biopsies were collected from early-onset AChR-MG patients with follicular hyperplastic thymus (*n* = 18), MG patients with thymoma (*n* = 30), and thymoma patients without MG (*n* = 33), who underwent thymectomy (Table 1). All MG patients, both those with follicular hyperplastic thymus and thymoma, had anti-AChR antibodies. No patient included in the study had other known neurological or autoimmune diseases. Non-MG thymoma patients did not have MG symptoms or other autoimmune diseases associated with thymoma, and were not treated with corticosteroids. As controls, normal thymuses from subjects without autoimmune diseases who underwent cardiovascular surgery (*n* = 10; 4 female and 6 male; age at surgery: 26.0 ± 16.1 years), and non-neoplastic normal thymic tissues adjacent to thymoma from non-MG thymoma patients (*n* = 5; 1 female and 4 male; age at thymectomy: 60.1 ± 12.0 years), were included in the study. Thymectomy and thymus histological classification were performed at the Humanitas Clinical and Research Hospital, I.R.C.C.S. (Milan, Italy). WHO types A (4 MG and 3 non-MG), AB (6 MG and 10 non-MG), B1 (2 MG and 4 non-MG), B2 (5 MG and 1 non-MG), mixed B1/B2 (3 MG and 5 non-MG), B3 (8 MG and 5 non-MG), and mixed B2/B3 (2 MG and 5 non-MG) thymomas were included in the analyses. For each thymus, some fragments were fixed in 10% formalin for histopathological classification; other fragments were snap-frozen and stored at −80 °C pending molecular analyses. 

Available serum samples from 13 of the 33 thymoma patients without MG were subjected to ELISA for testing anti-AChR, -RYR1, and -TTN antibodies, as described below. Anti-RYR1 and -TTN antibodies were also assessed in serum from 10 of the 18 MG patients with thymic hyperplasia, and 17 of the 30 MG thymoma patients. 

Serum samples from 14 healthy donors (8 female and 6 male; age at blood collection: 44.6 ± 10.4 years) and 27 MG patients (21 female and 6 male; age at blood collection: 43.8 ± 15.3) were subjected to ELISA for anti-human HSP60 antibody assessment, as described below. The 27 MG patients included 13 pre-thymectomy AChR-positive patients (9 female and 4 male; 6 with follicular hyperplasia and 7 with thymoma; age at blood collection 36.3 ± 15.4 years) and 14 seronegative (AChR-, MuSK-, and LRP4-negative) MG patients (12 female and 2 male; age at blood collection: 52.6 ± 9.7 years). 

The study was approved by the Fondazione IRCCS Istituto Neurologico Carlo Besta Ethics Committee (protocol code: 586/2014). Patients and controls signed an informed consent form for using their biological samples for research.

### 5.2. RNA Isolation

Total RNA was extracted from frozen thymic fragments using TRIzol method, according to manufacturers’ instructions (Thermo Fisher Scientific, Waltham, MA, USA). Quality and concentration of RNA were evaluated by NanoDrop 2000 spectrophotometer (Thermo Fisher Scientific).

### 5.3. Reverse Transcription and Real-Time PCR

Total RNA samples were reverse transcribed using Superscript VILO cDNA synthesis kit (Thermo Fisher Scientific). cDNAs were amplified by quantitative real-time PCR, using predesigned functionally tested TaqMan gene expression assays specific for *CHRNA1*, *RYR1*, *TTN*, *NEFM*, *RYR3*, *HSP60*, and *AIRE*, on the ViiA7 real-time PCR system (Thermo Fisher Scientific). Human 18S was used as endogenous control for normalization of gene expression data. Transcriptional levels of target genes were expressed as relative values normalized with 18S, using the formula 2^−ΔCt^ × 100.

### 5.4. Quantification of Anti-AChR, -RYR1, -TTN, and -HSP60 Antibodies 

Autoantibodies to AChR were assessed using a commercial AChR-Ab radioimmunoprecipitation assay (RIA) kit (RSR Limited, Cardiff, UK). Anti-RYR1 and -TTN antibodies were tested using a homemade Western Blot based on RYR1 purified from rabbit muscles, and by a commercial enzyme-linked immunosorbent assay (ELISA) kit (DLD Diagnostika GmbH, Hamburg, Germany), respectively.

Antibodies to human HSP60 were quantified in serum from 27 AChR-positive and 19 seronegative MG patients and 14 healthy controls, using the Human Hsp60 Autoantibody ELISA Kit (Novus biologicals, Bio-techne SRL, Milan, Italy).

### 5.5. Statistical Analysis

The non-parametric distributed data, tested via Shapiro–Wilk test, were analyzed with Mann–Whitney test for comparison of two groups. Differences were considered statistically significant when the *p* values were <0.05. Pearson’s correlation coefficient was measured for correlation analyses. Heatmap of log2 transformed normalized gene expression values (log2(2^−ΔCT^ × 100)) was obtained using R statistical package. GraphPad Prism v 8.0 (La Jolla, San Diego, CA, USA) was used for data elaboration and statistical analyses.

## Figures and Tables

**Figure 1 biomedicines-11-00732-f001:**
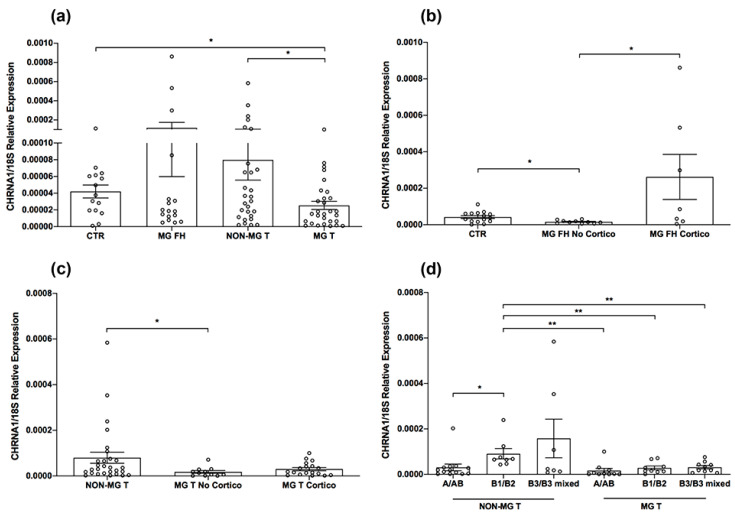
*CHRNA1* expression in MG and control thymuses. (**a**) *CHRNA1* mRNA levels were grouped as follows: normal thymuses (*n* = 15, CTR); follicular hyperplastic MG thymuses from both corticosteroid-naïve and -treated patients (*n* = 18, MG FH); thymomas from non-MG patients (*n* = 33, NON-MG T); and thymoma from both corticosteroid-naïve and -treated MG patients (*n* = 30, MG T). (**b**) *CHRNA1* mRNA levels in MG FH thymuses from patients stratified in corticosteroid-naïve (MG FH No Cortico) and -treated (MG FH Cortico), according with corticosteroid treatment before thymectomy, compared to CTR. (**c**) *CHRNA1* mRNA levels in thymomas from corticosteroid-naïve (MG T No Cortico) and -treated (MG T Cortico) patients, compared to NON-MG T. (**d**) *CHRNA1* mRNA levels in NON-MG T and MG T stratified according with WHO thymoma type. In the graphs, *CHRNA1* expression levels were expressed as relative values (2^−ΔCt^ × 100) that were normalized toward the endogenous 18S. Mann–Whitney test, * *p* < 0.05; ** *p* < 0.01.

**Figure 2 biomedicines-11-00732-f002:**
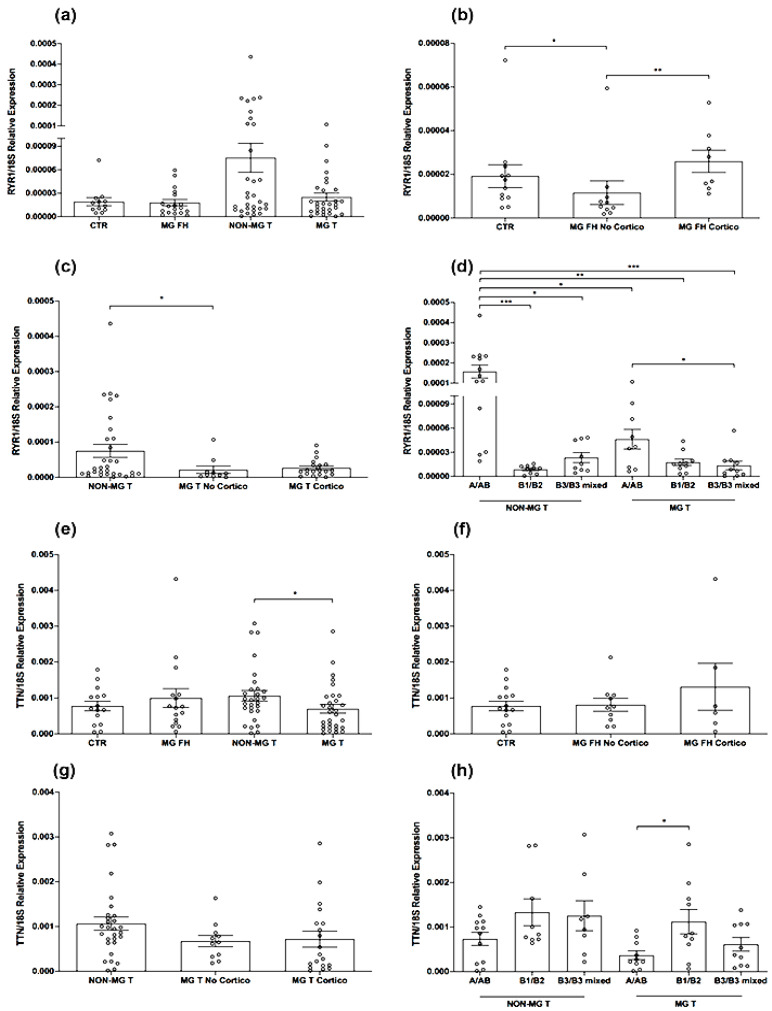
*RYR1* and *TTN* expression in MG and control thymuses. (**a**) *RYR1* mRNA levels in normal thymuses (CTR), follicular hyperplastic MG (MG FH) thymuses, thymomas from non-MG (NON-MG T), and MG (MG T) patients. (**b**) *RYR1* mRNA levels in MG FH thymuses from corticosteroid-naïve (MG FH No Cortico) and -treated (MG FH Cortico) patients compared to CTR. (**c**) *RYR1* mRNA levels in MG thymomas from corticosteroid-naïve (MG T No Cortico) and -treated (MG T Cortico) patients compared to NON-MG T. (**d**) *RYR1* mRNA levels in NON-MG T and MG T stratified according with WHO thymoma type. (**e**) *TTN* mRNA levels in CTR, MG FH, NON-MG T, and MG T. (**f**) *TTN* mRNA levels in MG FH No Cortico and Cortico, compared to CTR. (**g**) *TTN* mRNA levels in MG T No Cortico and Cortico, compared to NON-MG T. (**h**) *TTN* mRNA levels in NON-MG T and MG T stratified according with WHO thymoma type. In the graphs, *RYR1* and *TTN* expression levels were expressed as relative values (2^−ΔCt^ × 100) normalized toward the endogenous 18S. Mann–Whitney test, * *p* < 0.05; ** *p* < 0.01; *** *p* < 0.001.

**Figure 3 biomedicines-11-00732-f003:**
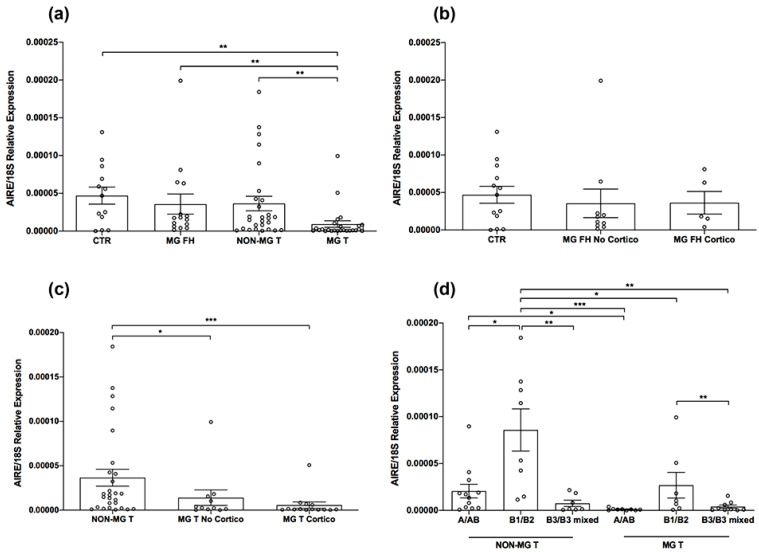
*AIRE* expression in MG and control thymuses. (**a**) *AIRE* mRNA levels were assessed in normal thymuses (CTR), follicular hyperplastic MG (MG FH) thymuses, thymomas from non-MG patients (NON-MG T), and MG thymomas (MG T). (**b**) *AIRE* mRNA levels in MG FH thymuses from corticosteroid-naïve (MG FH No Cortico) and -treated (MG FH Cortico) patients compared to CTR. (**c**) *AIRE* mRNA levels in MG T from corticosteroid-naïve (MG T No Ccortico) and -treated (MG T Cortico) patients compared to NON-MG T. (**d**) *AIRE* mRNA levels in NON-MG T and MG T stratified according with WHO thymoma type. In the graphs, *AIRE* expression levels were expressed as relative values (2^−ΔCt^ × 100) normalized toward the endogenous 18S. Mann–Whitney test, * *p* < 0.05; ** *p* < 0.01; and *** *p* < 0.001.

**Figure 4 biomedicines-11-00732-f004:**
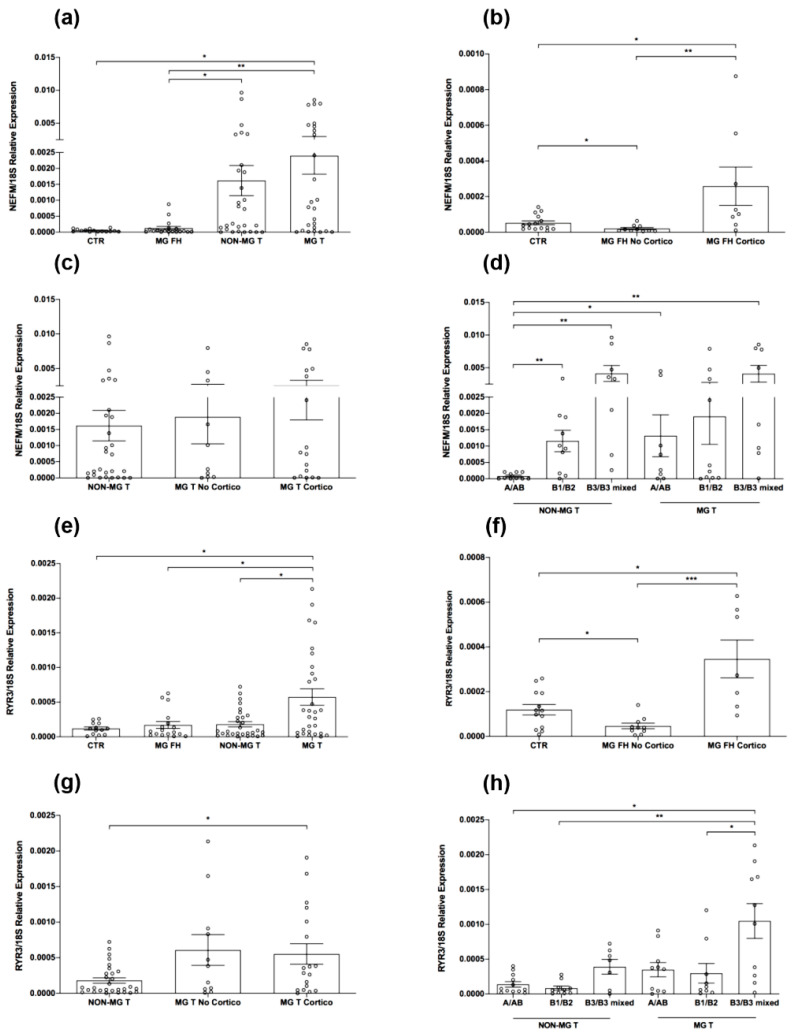
*NEFM* and *RYR3* expression in MG and control thymuses. (**a**–**d**) *NEFM* mRNA levels in normal thymuses (CTR), follicular hyperplastic (MG FH) thymuses from MG patients stratified in corticosteroid-naïve (MG FH No Cortico) and -treated (MG FH Cortico), thymomas from non-MG patients (NON-MG T), thymomas (MG T) from MG patients stratified in corticosteroid-naïve (MG T No cortico) and -treated (MG T Cortico), and NON-MG T and MG T stratified according with WHO thymoma type. (**e**–**h**) *RYR3* mRNA levels in CTR thymuses, MG FH from patients stratified in MG FH No Cortico and MG FH Cortico, NON-MG T and MG T from patients stratified in MG T No Cortico and MG T Cortico, and non-MG T and MG T stratified according with WHO thymoma type. In the graphs, gene expression levels were expressed as relative values (2^−ΔCt^ × 100) that were normalized toward the endogenous 18S. Mann–Whitney test, * *p* < 0.05; ** *p* < 0.01; and *** *p* < 0.001.

**Figure 5 biomedicines-11-00732-f005:**
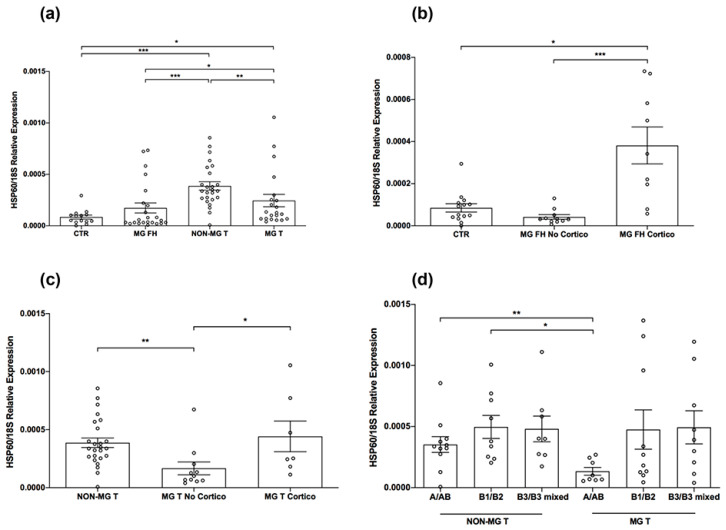
*HSP60* expression in MG and control thymuses. (**a**–**d**) *HSP60* mRNA levels in normal thymuses (CTR), follicular hyperplastic (MG FH) thymuses from MG patients stratified in corticosteroid-naïve (MG FH No Cortico) and -treated (MG FH Cortico), thymomas from non-MG patients (NON-MG T), thymomas (MG T) from MG patients stratified in corticosteroid-naïve (MG T No Cortico) and -treated (MG T Cortico), and NON-MG T and MG T stratified according with WHO thymoma type. In the graphs, gene expression levels were expressed as relative values (2^−ΔCt^ × 100) normalized toward the endogenous 18S. Mann–Whitney test, * *p* < 0.05; ** *p* < 0.01; and *** *p* < 0.001.

**Figure 6 biomedicines-11-00732-f006:**
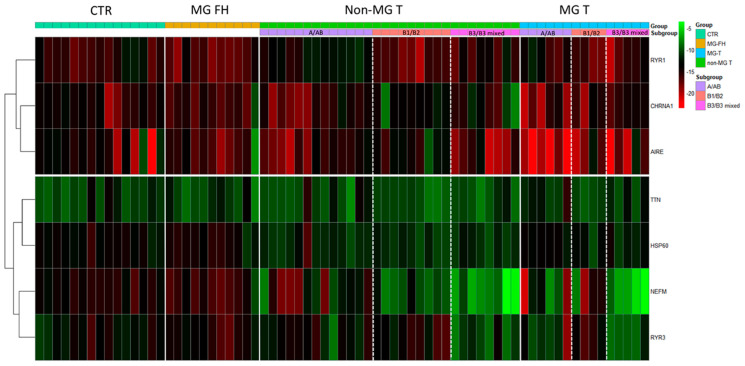
Heatmap of log2 transformed normalized gene expression data (log2(2^−ΔCT^ × 100)). The columns (samples) are unclustered, whereas the rows (genes) are clustered according to differential gene expression in MG compared to normal control thymuses (CTR) and thymomas from non-MG patients (NON-MG T). MG thymuses included follicular hyperplastic (MG FH) and thymoma (MG T) thymuses from corticosteroid-naïve MG patients. The color and intensity of the boxes are used to represent relative values of gene expression, with red representing lower levels and green representing higher levels.

**Figure 7 biomedicines-11-00732-f007:**
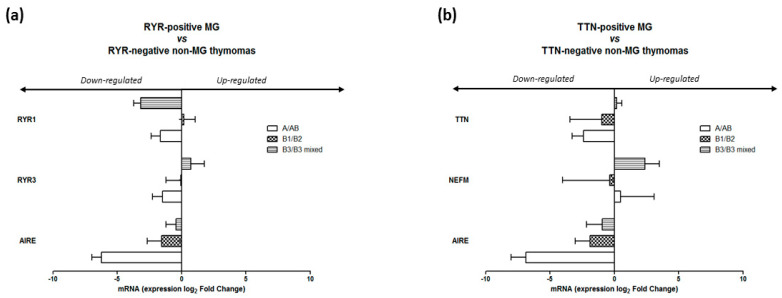
Altered expression of *RYR1, RYR3*, and *AIRE,* and of *TTN, NEFM*, and *AIRE* in thymoma from MG patients positive for anti-RYR1 (**a**) and -TTN (**b**) autoantibodies compared to thymoma from seronegative non-MG patients. Data are presented as mean ± SEM of log2-fold change ratios of 2^−ΔCt^ × 100 expression values in MG thymomas grouped based on WHO subtypes versus the same subtypes in the non-MG thymoma group.

**Table 1 biomedicines-11-00732-t001:** Main features of MG patients and controls included in the study.

	Control Thymuses (*n* = 15) ^1^	Hyperplastic MG Thymuses (*n* = 18)	Non-MG Thymomas (*n* = 33)	MG Thymomas (*n* = 30)
Sex (F:M)	5:10	15:3	11:22	14:16
EOMG:LOMG ^2^	-	18:0	-	10:20
Age at surgery (years, mean ± SD)	43.5 ± 24.11 ^3^	28.61 ± 8.32	57.42 ± 13.65	51.85 ± 11.94
Corticosteroid-treated patients	None	8	None	10
WHO types A AB B1 B2 B1/B2 mixed B3 B2/B3 mixed	-	-	3 10 4 1 5 5 5	4 6 2 5 3 8 2

^1^ Control thymuses were normal thymuses from 10 cardiopathic patients without autoimmune disorders (4 female and 6 male) and normal tissues adjacent to thymoma from 5 non-MG thymoma patients (1 female and 4 male). ^2^ EOMG: MG with early onset (<50 years); LOMG: MG with late onset (>50 years). ^3^ Age at cardiovascular surgery of the 10 cardiopathic patients was 26.0 ± 16.1 years; age at thymectomy of the 5 non-MG thymoma patients was 60.1 ± 12.0 years.

## Data Availability

The data presented in this study are available on request from the corresponding author, P.C.

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
