# Peer review of "Muscle and Muscle-like Autoantigen Expression in Myasthenia Gravis Thymus: Possible Molecular Hint for Autosensitization"

_biomedicines, 2023, doi:10.3390/biomedicines11030732_

Round 1

Reviewer 1 Report

General comments

In their manuscript, Iacomino et al. investigated mRNA expression for muscle antigens (CHRNA1, RYR1, and TTN), muscle-like antigens (NEFM, RYR3 and HSP60) and AIRE in the thymus of MG (thymoma or not associated MG) and non-MG thymomas patients, as well as controls.

Based on changes in the level of thymic expression of these molecules, they hypothesize that altered muscle antigen expression may favor autosensitization in MG thymus, and that molecular mimicry involving tumor-related muscle-like proteins may be a mechanism making thymoma prone to develop MG.

The manuscript is well written but might benefit from a few precisions.

Introduction:

The term “whole AChR” is confusing. TECs and myoid cells express all AChR subunits but in TECs the receptor is not functional (Siara et al. Neurology 1991).

Are the other muscle or muscle-like antigens expressed by TECs or myoid cells?

Are the other genes investigated in this study known to be controlled by AIRE? Or by other regulators of tissue-specific antigens such as Fezf2 or Prdm1?

Materials and Methods:

For control thymuses, the authors have mixed thymic tissues from 10 donors with cardiovascular diseases (mean age 26 years old) and 5 donors with non-MG thymoma using the non-neoplastic adjacent tissue (mean age 60 years old). Are there differences in these 2 subgroups of donors for the expression of the investigated genes? This should be checked.

 Specify clearly if thymoma patients without MG were not treated with corticosteroids.

 In each figure legend, indicate the statistical test that has been used as in the methods sections, two different tests are proposed (Mann Whitney or Kruskal-Wallis).

 Results

The authors investigated the expression of muscle or muscle-like antigens in whole thymic fragments. These thymic fragments contain various cells in different proportions depending on the pathogenic thymus, such as fewer myoid cells in thymoma but also non-thymoma MG thymus. The decrease in muscle or muscle-like antigen mRNA expression could be correlated with a decrease in myoid cells. The expression of a specific muscle/myoid gene (not considered as an autoantigen) could be investigated to exclude this possibility.

For the last paragraph, figure 7 is difficult to understand, the title above each graph is confusing. The text on the figure should be written bigger.

I do not understand the conclusion. “These results suggest that down-regulation of muscle antigens” : this is fine only for A/AB patients “up-regulation of muscle-like tumor antigens” the up regulation is not so clear” .

A/AB patients with autoantibodies against RYR1 tend to have a lower thymoma expression of AIRE and RYR1. Is there a correlation between AIRE and RYR1 mRNA expression? Is it specific of RYR1 and what about TTN (the other muscle antigen)?

The same is observed for A/AB patients with autoantibodies against TTN that tend to have a lower thymoma expression of AIRE and TTN. So, is it specific of TTN and what about RYR1 (the other muscle antigen)?

 Discussion:

The authors do not discuss the fact that changes in terms of global mRNA expression could also reflect changes in the proportion of specific cells. Such as myoid cells as mentioned above or even B cells. AIRE is known to be expressed in B cells could this affect the global expression of AIRE in hyperplastic thymuses and hide a decrease in TECs?

Reviewer 2 Report

The paper presented to me for review provides new insights into the intrathymic alterations underlying autosensitization to muscle autoantigens in AChR-MG patients. The work undoubtedly contributes new insights into the molecular knowledge of this disease. The paper is written in a typical, clear and lucid manner. A rather large group of patients was analyzed, which provides reliable observations. 

However, the paper requires several additions and clarifications before it can be accepted for publication:

1. Whether the patients enrolled in the study (in all groups) had other neurological, autoimmune, rheumatic, etc. diseases. This was not clearly indicated in the methodology and, of course, such diseases can affect the results. 

2. In the introduction regarding myasthenia gravis, the information should be completed that this is a disease that most likely also occupies the central nervous system as shown in the paper: https://pubmed.ncbi.nlm.nih.gov/34439676/

3. References from before 2018 should be reviewed and leave only those items that bring the necessary insight into the topic.
